# MRI Methods to Visualize and Quantify Adipose Tissue in Health and Disease

**DOI:** 10.3390/biomedicines11123179

**Published:** 2023-11-29

**Authors:** Katerina Nikiforaki, Kostas Marias

**Affiliations:** 1Computational BioMedicine Laboratory, Institute of Computer Science, Foundation for Research and Technology—Hellas, 70013 Heraklion, Greece; kmarias@ics.forth.gr; 2Department of Electrical and Computer Engineering, Hellenic Mediterranean University, 71410 Heraklion, Greece

**Keywords:** MRI, MRS, adipose tissue, relaxometry, fat fraction, NAFLD, NASH, steatosis

## Abstract

MRI is the modality of choice for a vast range of pathologies but also a sensitive probe into human physiology and tissue function. For this reason, several methodologies have been developed and continuously evolve in order to non-invasively monitor underlying phenomena in human adipose tissue that were difficult to assess in the past through visual inspection of standard imaging modalities. To this end, this work describes the imaging methodologies used in medical practice and lists the most important quantitative markers related to adipose tissue physiology and pathology that are currently supporting diagnosis, longitudinal evaluation and patient management decisions. The underlying physical principles and the resulting markers are presented and associated with frequently encountered pathologies in radiology in order to set the frame of the ability of MRI to reveal the complex role of adipose tissue, not as an inert tissue but as an active endocrine organ.

## 1. Introduction

Magnetic Resonance Imaging (MRI) is the modality of choice for soft tissue imaging in a very wide range of pathological entities, exhibiting a number of advantages over other imaging methods, such as the lack of ionizing radiation, the ability of the user to acquire images at any plane based on anatomy and scanners being readily available and at reasonable cost. Moreover, MRI can deploy contrast mechanisms providing information from different scales of structure, beginning from the molecular and reaching to the macroscopic level, at a single acquisition session. Apart from pure image contrast perceived as variations in signal intensity, contemporarily used MR methodologies decipher the derived information into robust and repeatable biomarkers correlated with different aspects of pathology or physiology status, aiding thus diagnosis, disease classification, and selection of optimal strategy for patient management related to therapy response or planning. In recent decades, the prevailing notion of functionality restricted to storing energy and passively acting as a natural insulator has been thoroughly reconsidered. Modern physiologists now identify the significance of adipose tissue not inferior to that of an endocrine organ, participating and regulating metabolic processes. This fact totally reshapes the role of MRI in clinical imaging as conventional techniques have to be revised to meet the current needs and emerging techniques have to be made adequately sensitive to provide insight at multiples levels of granularity regarding the role of adipose tissue.

## 2. Lipid and Water Molecules in a Magnetic Field

MRI signals originate mostly from protons (unpaired spins) in the fat and water molecules and, more specifically, from hydrogen nuclei in water molecules or fatty acid chains composing adipose tissue. These are the main signal sources in MR imaging and, luckily, they are very abundant in the human body. However, the nature of water and fat molecule is responsible for the very distinct behavior of each one, which in turn is the basis for the ability to selectively image, amplify or suppress the signal of one of the two and importantly quantify, depending on the clinical need for a certain referral to the scanner suite.

The structure and function of adipose tissue varies by location, including subcutaneous white and brown fat, visceral fat, bone marrow fat, and fat within organs or muscle. Fat, as opposed to water, is a slowly moving large molecule. It exists in tissues primarily as long chain of triglycerides, and most of their hydrogen atoms are nestled in aliphatic side chains among relatively electroneutral carbon atoms. Water, on the other hand, is a small molecule with two hydrogen and one oxygen atom. It is the main constituent of human body, making up almost 80% of the muscle tissue, more than 70% of the brain, and even more than 30% of the bones. The difference of the two molecules in size and ability to move, as well as the type of chemical bonds within the molecules, affect their behavior during the MR acquisition process.

Water is a rapidly rotating molecule, with a polar character due to its electronegative oxygen atom. Hydrogen atoms are bond to the water molecule by strongly polar bonds, and this polar nature of the atom imposes a certain symmetry rendering the H nuclei equivalent with respect to their environment. The strongly electronegative oxygen atom of a water molecule pulls away the protective electron clouds covering the hydrogen nuclei. This deshielding effect exposes the hydrogen protons to a relatively stronger local magnetic field than they would otherwise experience. Hence, they resonate slightly faster than the more shielded protons in triglyceride molecules. This difference in resonance frequencies between water and fat is named chemical shift, and is the physical principle behind many MR methodologies separating fat from water signal based on their resonance frequency.

These differences are more prominent among different molecules, but subtle changes exist within the fat molecule itself among hydrogen atoms, constituting a more complex spectrum for fat, which is composed of a number of different resonance frequencies with known shifts among them in the fat spectrum. They arise from the local differences caused by the variable shielding degree of each hydrogen in its very local chemical environment in the vicinity of the chemical bond. The resulting spectrum from all H nuclei present in the human body comprises a single resonance frequency for the water molecule and a number of different resonance frequencies from adipose tissue, with amplitudes relative to their concentration. This spectrum is the human fingerprint in each elementary volume in the glossary that an MRI scanner comprehends.

## 3. MRI Methodologies

The two main parts of every MRI process is the initial excitation with the RF pulses and the progression of a relaxation to the initial state immediately after, where the signal from the two pools is recorded. While relaxing, H nuclei protons emit energy, which is then received as electrical signal by coils placed adjacent to the body area of interest. The contributions from the fat and water depots can be distinguished both spatially, from which part of the body they come from in order to be placed to the corresponding part of the body, but also chemically, i.e., to identify its origin, either from fat or water. MR contrast stems from finely adjusting acquisition parameters to preferentially amplify the contrast indicative of the physical phenomenon of interest over a number of other physical phenomena that evolve concurrently and also affect the resulting signal.

The main characteristics differentiating fat and water signal can be summarized in:The fat molecule, being large and ponderous, is the first to lose the energy deposited by RF pulses by interacting with its environment and returns faster than the water molecule to its initial state. This fact is measurably registered in two biomarkers called T1 or T2 relaxation constant (described in more detail in Section 3.1) capturing the rate at which each tissue type returns to its initial equilibrium status, before the RF pulse. Once the total amount of energy is expended to interactions with the environment, the nuclei are invisible to the receiver coils and thus the imaging process.As mentioned before, the MR signal comes from the single water and the multiple fat resonance peaks. Water and fat are identified by the scanner via their known position(s) in the spectrum. For a number of applications, it suffices to assume that the fat spectrum comprises only the dominant one, while for other applications the relative amplitudes of the fat spectrum peaks are the main outcomes of the imaging session.

The applied RF pulses can be designed either to excite all hydrogen protons of a certain body thick body volume, of a thin slice or even of smaller elementary volume (voxel The pattern of relaxation depends on the magnetic properties of each particle and the net effect of all particles’ contribution within a voxel determines the contrast of the image for this specific anatomical site. The rate at which each proton relaxes to the initial state before the application of the RF pulse sequence differs. Consequently, when sampling the available signal of a certain location at a certain time point during signal readouta different amount of signal is assigned to each voxel, depending on the content of the anatomical site in adipose or aqueous protons.

The sum of all signals is processed and each contribution is mapped to the body location that it originates from to create the final image. Apart from the two indispensable parts of any MRI acquisition scheme (excitation and relaxation), many variations exist regarding the design of the rest part of the imaging sequence, such as the RF pulse design and ordering and the reading process of the emitted signal. They aim to accentuate in terms of signal and then visually present a different property of the resonating nucleus. These characteristics may be derived from different scales of body structure, even significantly smaller that the picture elements that are recorded on the image. The MRI armamentarium comprises numerous applications either creating anatomic representations of human organs or quantitatively estimating tissue properties to respond to certain clinical needs. The most widely used are discussed herein, focusing on applications concerning the physiology and pathology related to adipose tissue.

### 3.1. Relaxometry

Relaxometry relies on observing by consecutive series of imaging the process of the nucleus returning in its initial state through consequent acquisitions in time. This method provides a representation of the evolution of the magnetization vector for any elementary volume in time complete relaxation where the MR signal reaches noise levels. Relaxometry sequences have the advantage of providing a tissue specific result and not being dependent on acquisition parameters such as the specific details of the MR sequence. The result is a quantitative marker of the relaxation rate at which the nuclei lose the excess energy deposited after the excitation pulse. These rates are measured independently in the longitudinal axis, i.e., parallel to the external magnetic field (T1 relaxometry) and perpendicular to it (T2 relaxometry). The reason for this distinction is that those relaxation processes are governed by different physical phenomena, related to the mechanism of energy loss to their environment.

Relaxometric sequences date back to the first days of MRI when Damadian, in his emblematic paper [1], discussed the role of relaxometry in detecting cancer. The basis of these techniques is that each imaging object bears a distinct identity with respect to its magnetic properties. Thus, any deviation from the expected values increase suspicion for pathology or disease. As the T1 and T2 relaxation constants of fat are significantly lower than that of the water, deviation from the expected values within the parenchyma of a certain organ of high aqueous concentration are indicative of changes in its composition, possibly presence of fat. The presence of non-macroscopic fat components that only partially occupy imaging volume are not easily identified by the visual inspection of the image by the expert, if they exist in low concentrations; however, they affect the overall relaxation pattern within this voxel to a degree related to their abundance. Observing the relaxation process of that voxel in detail can unmask the presence of fatty component by observing the pattern of relaxation in time. The pattern can be extracted and observed at a very local (pixel-based) level throughout the abdominal organ without suffering from sampling errors. Simple post-processing actions readily available in almost all scanner quantify the pattern into relaxation rates measured in time units, most frequently milliseconds. For adipose tissue specifically, T1 and T2 relaxation constants are approximately two hundred milliseconds, with T1 always being longer than T2 for any tissue. Comparably, water molecules exhibit T1 and T2 relaxation rates approximately one order of magnitude slower, i.e., above 1 s. Applications are numerous, as relaxometry techniques can characterize tissue properties independently of the imaging protocol used.

*T1 relaxometry*: T1 relaxation has been studied as a major component of a descriptive index for detecting accumulation of fat in abdominal organs such as in non-alcoholic fatty steatohepatitis (NASH), which can progress into a more serious pathologic condition, non-alcoholic fatty liver disease (NAFLD). It can be also combined with other MR-based techniques such as fat fraction estimation to create a more thorough disease profile as complementary information integrated into an index of disease severity [2].

*Identification of microscopic fatty components*: Apart from calculating the overall relaxation rate within a voxel, it is possible to assume more than one relaxing population within the same voxel and therefore decompose the complex pattern of relaxation into individual components, one relaxing faster than the other. This technique can also reveal the participation of an adipocytic component for a structure of unknown composition as evidence for its origin, as for example in complex soft tissue masses [3,4].

### 3.2. Conventional T1 and T2 Contrast

Relaxometry, notwithstanding its specificity, has not gained an accordingly crucial place in every day practice. The main reason is that it is a time consuming process for clinical imaging time slots. On the contrary, conventional T1 or T2 weighted images are the main parts of every examination being in essence snapshots of the relaxation process at a chosen time point. The snapshot timing is chosen to capture the status of the magnetization vector either optimally to highlight the relaxation status concerning the transverse magnetization component (T2 weighted) or the longitudinal one (T1 weighted) in one time frame of the signal evolution to complete proton relaxation.

In practice, this selection of time point and thus T1 or T2 weighting is performed by user-defined parameters which favor the contrast (relative difference in the acquired signal) between two tissue types in order to make them distinguishable visually. Thus, each structure is identified on the basis of its grey level in the final image. Such techniques are characterized as “weighted”, highlighting the fact that both T1 or T2 phenomena contribute to the contrast but one of them is rendered dominant compared to the other by the appropriate selection of acquisition parameters. Both in T1 and T2 weighted sequences fat appears to be bright, while it is sometimes selectively suppressed by the user in order to highlight other structures bearing similar contrast, as fluids or edema in T2 images to increase disease conspicuity. T1 and T2 images can be obtained from any anatomic part and can be used to create tissue representations at any plane of orientation at a desired resolution and image thickness. Applications are numerous and concern the whole spectrum of medical conditions.

*Recognition of fatty tissue*: Intuitively, identifying fat depots based on the expected contrast in both T1 and T2 images serves the purposes of identifying physiological and pathological structures in the whole continuum of diagnostic images. Ectopic fat at larger aggregations can be localized as extramyocellular or inramyocellular lipids are frequently based on T1 weighted images as fat is visible as bright signal among low signal muscle tissue in the same acquisition. Also, body parts deprived of fat can be diagnosed on conventional T1 or T2 images, such as cachexia or sarcopenia [5] It has to be noted that T1 is usually preferred over T2, as in the latter bright fat signal is similar to bright fluid signal.

*Oncologic applications*: Oncologic differential diagnosis also benefits from the identification of fatty components in conventional T1 or T2 imaging, since the imaging features of each one are characteristic and correlate with the histologic profile of the masses [6]. MRI scans can reveal the presence of septa, the signal homogeneity across the lesion volume, and the co-existence of cystic compartments within adipocytic tumors among other characteristics. Moreover, T1 acquisitions after administration of contrast medium are used to recognize the pattern of enhancement, which is informative of the vascular elements of the mass and its neighboring area. One such example in oncology is differentiation of lesions typically containing fat such as adrenal adenomas or angiomyolipomas, and support differential diagnosis from other entities that do not have fatty components, i.e., carcinomas and metastases [7].

*MSK applications*: Many T2 weighted acquisitions, especially for MSK applications, are modified to suppress one of the two signal pools to verify the presence of the other, such as fat suppressed or STIR acquisitions to detect bone marrow abnormalities. Areas of high signal intensity in T2 weighted images can be suspicious for either benign or malignant due to the higher free water content than the surrounding normal marrow adipose tissue (MAT).

*Fat volumetry*: Knowing the imaging parameters of conventional T1 or T2 images and specifically the voxel size, sequential tomographic MRI images are used to estimate the adipose tissue volume in various depots, mainly subcutaneous adipose tissue or visceral adipose tissue. This relative quantification can provide important information in metabolically healthy versus metabolically unhealthy obesity regardless of a healthy body mass index, where the ratio of the volume of visceral and subcutaneous adipocytes plays a significant role [8].

### 3.3. Fat Fraction

The chemical shift phenomenon between fat and water can be used to provide information on the intravoxel fat fraction of the body as a quantitative percentage of fat fraction (signal of fat over total signal). It is important to differentiate this technique from other acquisitions depicting macroscopic fat pools as the above-described T1 and T2 weighted images, since the importance of the result lies in the ability to reveal even minimal fatty infiltration not visually perceived in standard practices including breath-hold acquisitions. Moreover, as an accurate quantitative measurement, it can be used to detect even subtle changes after therapeutic interventions or for longitudinal disease progression monitoring.

Because of the chemical shift, intravoxel fat and water magnetization vectors become aligned or opposed with each other twice per cycle. When they are aligned the result determining the final contrast for that voxel arises from the addition of fat and water nuclei within this volume, while at the opposed phase, the final contrast is defined by the subtraction of the fat and water signal. Simple mathematical calculations between the two images, in-phase or out-of-phase, can provide a water-only image, a fat-only image or the ratio of either of the two over the total signal available from a certain voxel. This method, called Dixon’s Method [9], has been later modified to accommodate a larger number of images [10,11] in order to increase its accuracy in detecting the percentage of fat fraction. Moreover, the more recent techniques are also able to quantify fat fraction above 50%, which was a major limitation of the initial fat-water separation technique.

The technique is able to produce four different contrasts with the same data: water only, fat only, in-phase, and out-of-phase. The fat-only images offer the potential for fat-quantification, while the water only images can be used as fat suppressed images with the advantage over conventional fat suppressed images of robust and homogeneous fat suppression. The acquisition times are longer than conventional images, but shorter from relaxometric studies.

Applications are similar to the ones mentioned for fat detection with T1 or T2 fat suppressed contrast, adding the advantage of quantification to the diagnostic confidence. Separating fat and water images to calculate fat fraction rightly earned their position in a routine abdominal MR protocol when quantitative information is required to highlight spatial heterogeneity of fat accumulation within an organ, minor levels of fatty infiltration or in longitudinal studies for following up patients.

*Liver fat fraction*: Derangements in liver function results in abnormal storage of intrahepatic fat. NAFLD along with iron deposition in the liver are the most common causes of chronic liver disease, and suggest a feature of metabolic syndrome related to obesity, type 2 diabetes, and cardiovascular disease. Since the molecular mechanisms regarding the progression of health to fatty liver to NASH or to NAFLD remain poorly understood, a reliable metric offers insight potential surrogate marker to support diagnosis and provides feedback during or after any therapeutic intervention. Color coded voxel based fat fraction maps produced by fat-water separation techniques are a great tool to identify areas of isolated steatosis, assess disease severity and are also able to follow up disease progression [12]. It serves as an alternative to liver biopsy, sparing patients form risks and discomfort inherent of such interventions. It is noteworthy, that the co-existence of iron deposition in the liver parenchyma negatively affects the ability to accurately define the degree of steatosis due to the paramagnetic nature of iron significantly affecting the signal evolution process.

*Bone marrow fat fraction*: Bone marrow is a dynamic tissue primarily composed of hematopoietic red blood cells and yellow fatty marrow. MAT quantity and its composition are etiologically related to metabolic disorders, such as obesity and type 2 diabetes mellitus, which negatively affect bone health [13]. Higher marrow fat is related to lower bone density, osteoporosis and prevalent vertebral fracture [14,15]. Moreover, marrow fat content changes during physiological ageing, but also in pathological situations such as myelodysplastic syndromes or after therapeutic interventions such as radiotherapy, [16,17,18,19]. Fat fraction quantifications is a far more sensitive marker than conventional T1 or T2 imaging contrasts to detect the interference of fat and water signal and report clinically relevant changes.

### 3.4. Magnetic Resonance Spectroscopy (MRS)

Fatty acid chemical composition can be defined by Magnetic Resonance Spectroscopy (MRS). For the vast majority of MR methodologies, such as the ones previously presented herein, the whole extent of fat spectrum is not considered. Instead, a weighted average value in the range of most plentiful fat components is generally quoted for the sake of simplicity in imaging representations and also in calculations. In MR spectroscopy, however, the desideratum is exactly to reproduce each peak in the adipose tissue spectrum and quantify the relative quantities of each specific resonance frequency. MRS aims to characterize adipose tissue in terms of the number of double bonds, fraction of saturated, monounsaturated, and polyunsaturated fatty acids, or semi-quantitative unsaturation indices [20,21]. To extract those descriptive indices, it suffices to measure the amplitude of each distinct fat peak by spectroscopic MR sequences.

A fat molecule consists of two main components—glycerol, which is a small organic molecule with three hydroxyl groups and a long chain of hydrocarbons to which a carboxyl group is attached. Most commonly, the number of carbons in the fatty acid are 12–18 carbons. The most plentiful H nuclei in fat are those in methylene (−CH2−) and methyl (−CH3) groups, measured to be approximately 3.5 parts per million (420 Hz at 3Tesla scanners) distant from the water peak. Hydrogens in carboxyl groups are not equivalent as in the polar water molecule H, as the non-polar they experience slightly different magnetic field in their local microenvironment and therefore have differences in their resonance frequency. As a result, fat spectrum as opposed to that of the water is more complex and is usually approximated with a model of 6 or 9 peaks of different resonance frequencies and known relative shifts among them. The number of resolvable peaks depends on the magnetic field strength, its homogeneity, and the concentration of each molecular moiety.

It has to be noted, though, that spectroscopic techniques require expertise and special software and thus are not readily available. Moreover, they cannot be applied to extensive volumes in body and thus suffer from sampling errors. However, they provide a more detailed view of the adipocyte profile than the other methods discussed above for ectopic fat accumulation in the liver, bone marrow, or for the characterization of fat content in subcutaneous or abdominal fat depots.

*Adipocyte triglyceride profile*: The detailed chemical composition of subcutaneous or visceral fat has been linked to a number of pathological conditions, and is also closely correlated to dietary habits as obesity, insulin resistance, hepatic steatosis, osteoporosis. Obesity and metabolic disorders may, for example, be associated with an increased proportion of saturated fatty acids in the liver and bone marrow, while a lower degree of unsaturation has been linked both to reduced bone quality and type 2 diabetes [22,23,24]. Furthermore, the degree of unsaturation in adipose tissue may be affected differently in the vicinity of benign or malignant tumors [25].

## 4. Brown Adipose Tissue in MRI

Brown adipose tissue (BAT) is thermogenic tissue with reported beneficial effect in cardiometabolic health and weight management [26]. BAT exhibits different structure and functionality from white adipose tissue [27]. Several of those differences can become the basis of MRI contrast, and can be highlighted by appropriately tailored MRI protocols to detect BAT, which in adulthood is scarce in a small number of body regions. Such BAT attributes are mainly the presence of numerous mitochondria, the co-existence of a water component, and the presence of multilocular lipid droplets instead of a central unilocular one in white adipocytes [28].

The iron rich mitochondria are responsible for the brown color of these adipocytes. Concerning MRI, the paramagnetic nature of iron disturbs the local magnetic field and accelerates the process of T2 relaxation. T2 sequences adjusted by the correct choice of sequences to be more sensitive to such disturbances are named T2*. Both T2* weighted images and T2* relaxometry can reveal possible BAT depots if present, independently if they are activated or not during imaging. Moreover, BAT composition includes a large fraction of water which can be detected in tissue spectrum, as well as in relaxometry studies with multiple compartments [29], as previously discussed.

Other microstructural differences between white and brown adipocytes that translate into specific MR contrast is the ability of the water molecules to move freely in the intracellular space, diffusion weighted sequences (DWI). The DWI contrast principle relies on signal reduction in the occurrence of restricted water motion, achieved by the addition of two extra RF pulses that accelerate the dephasing of stationary or slowly moving spins to a degree defined by the user. The signal reduction is directly proportional to nuclei ability to diffuse, both in terms of velocity and range, and this is quantified in the DWI derived biomarker named apparent diffusion coefficient (ADC). The inner mitochondrial membrane is a physical barrier for the water to move, and this restricted water motion is reflected in the ADC measurement as a significantly lower value for BAT depots compared to white adipocyte regions. This quantitative measurement is another example of MRI providing microscopic level information. It has to be noted that the imaging of brown fat is not very frequent with MRI, since functional techniques such as PET [30] have the advantage of revealing metabolic information which is the key functional attribute of those adipocytes.

*Obesity and metabolic disorders*: Because of the potential activation of BAT through non-shivering thermogenesis, it has been deployed for addressing obesity and related comorbidities. Exposure to cold in combination with dietary intervention can initiate a process called “beiging” [31] of white adipocytes, and suggests the metabolic activation from inert to metabolically active, i.e., acquiring the functional characteristics of BAT [32,33,34]. Identifying the presence of brown or activated (beige) adipocytes is an active field of research for addressing obesity and metabolic disorders [35,36], but can also be deployed in health, in the frame of promoting healthier lifestyles through subtle lifestyle interventions [37,38].

*Lipomatous tumors*: One special case of lipomatous tumors are hibernomas, which are benign masses with high brown fat concentration. As such, they can be differentiated from other masses based on the characteristic phenotypes of brown fat imaging [39]. Figure 1 presents the imaging phenotype of lower limb hibernoma, exhibiting different characteristics from subcutaneous or bone marrow fat. The presence of water is evident from the high signal in T2 images, as well as for the persistence of high signals even after fat suppression. Moreover, calculated T2 values are higher and the fat fraction is significantly lower in the whole area of the mass. The software used for T2 calculation is an in-house built software platform [40], while the fat fraction map was calculated by Evorad 2.1 software Tesla QMRI Utilities-X implemented on the Evorad^®^ (IKnowHealth, Athens, Greece) research PACS client platform.

## 5. Limitations

Notwithstanding the importance of adipose tissue imaging and quantification with respect to a clinical condition, there are certain limitations and concerns regarding that specific use of MRI in clinical practice. Firstly, although MRI imaging is a tool to study the volume and distribution of body fatty depots, the MRI scanning suite is not always comfortable for obese individuals. For the sake of main magnetic field inhomogeneity, clinical MRI scanners have a limited space for the patient, restricted to cylindrical bore of 60–70 cm diameter. Moreover, receiver coils have to be placed adjacent to the body structure of interest and their rigid form can be very restrictive for larger size individuals. In such cases, other coil options are available; however, this requires a level of expertise from the side of the technologist to avoid compromised signal to noise ratio or degraded image quality.

Furthermore, in rapid acquisition schemes, the relative chemical shift between fat and water results in misregistration of fat signal with respect to the anatomy. For example, the fat signal might be registered a number of pixels away from the true anatomical site where the signal originated from, as the scanner translates water resonance frequency into position in the image miscalculating the fat signal spatial origin. The actual distance of fat–water signal misregistration depends on acquisition parameters, such as the bandwidth per pixel and the pixel size. For some applications, fat signal suppression is the default acquisition setting, i.e., in rapid echo planar imaging, to avoid a fat-only image shifted and superimposed on a water-only image of the patient.

However, fat signal suppression might be necessary not only to avoid artifacts but also because the high fat signal obscures other pathologic entities of equal or similarly bright representation in the image. As main magnetic field inhomogeneity is a sine qua non condition for good imaging quality, any factor affecting homogeneity may cause fat suppression to be inhomogeneous or fail, possibly depriving diagnostic value from the images. Such factors can be metallic implants, air-tissue interfaces or hemorrhages. That is a common concern in cases where contrast medium is administered (increases signal in T1 weighted images) or when edema is present in the region of interest (high signal in T2 images) and incomplete fat suppression may decrease pathology conspicuity.

## 6. Conclusions

To properly image the endocrine, metabolic and hematological functions of fat, it is necessary for imaging methods to evolve and become sensitive enough to characterize relevant pathology and physiology in a manner that is repeatable accurate and robust. MR methodologies and related biomarkers can play a very important role in presenting macroscopic fat depots, in detecting microscopic fatty infiltration and in describing molecular fat content. The field of related applications is vast, ranging from metabolic disorders to tissue characterization. While there is an overlap among different MRI methodologies in the clinical applications addressed, the choice is on the clinical expert depending on the granularity of information required and the time, software and hardware constraints. To this end, it is critical that clinicians become familiar with such novel MRI paradigms to image and quantify adipose tissues in order to promote patient diagnosis and management.

## Figures and Tables

**Figure 1 biomedicines-11-03179-f001:**
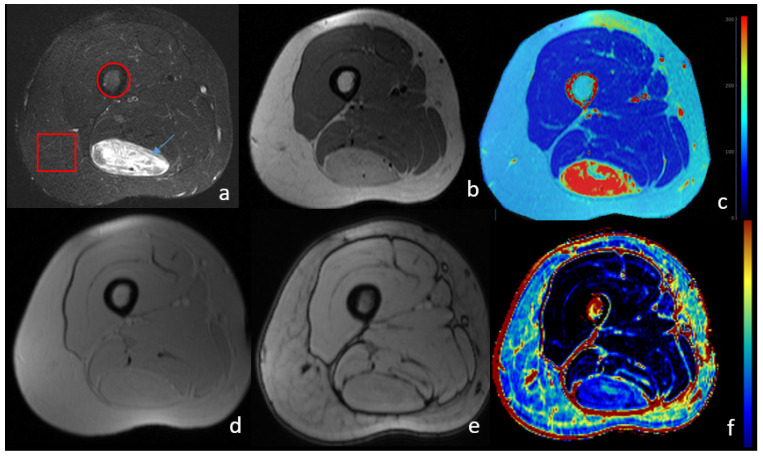
MRI image of lower limb with a brown fat lipoma (hibernoma). Histopathologic examination showed 90% of brown fat composition. The arrow shows the hibernoma mass, while the circle and square denote the bone marrow area subcutaneous fat depot, respectively. (**a**) T1-weighted with fat suppression; (**b**) T2- weighted; (**c**) T2 relaxometry parametric map calculated from multi echo T2 images. Brown fat lipoma exhibits high T2 relaxation rates due to the water component of the brown fat composition; (**d**) T1 in-phase; (**e**) T1 out-of-phase; (**f**) fat fraction map calculated from in- and out-of-phase images. Muscle shows minimal fat fraction, while a higher degree is observed for the bone marrow. Hibernoma exhibits smaller fat fraction, which is in accordance to the histopathologic examination showing BAT dominance over white adipocytes. Subcutaneous fat appears with lower fat fraction than the bone marrow as the technique is unable to quantify percentages of more than 50% fat dominance.

## Data Availability

Not applicable.

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
