# Peer review of "MRI Methods to Visualize and Quantify Adipose Tissue in Health and Disease"

_biomedicines, 2023, doi:10.3390/biomedicines11123179_

Round 1

Reviewer 1 Report

Comments and Suggestions for Authors Nikiforaki and Marias summarize in this review analyze the current status in fat tissue imaging with MRI, both physiological and pathological, and the application of these in quantitative biomarkers for adipose tissue pathology. The article gives a status of the art on a field that has only recently boomed given the increasing role that adipose tissue has acquired as an endocrine organ. The authors have shown an extensive review of the current literature on the topic, including providing an adequate bibliography on the subject.
Here are some minor topics to consider in subsequent versions of the review:
1. Page 1; paragraph 1 “Introduction”: here is stated about MRI “…the ability of the user to acquire images at any plane based on anatomy and scanners being readily available and relatively cheap”. The authors imply that MRI is easily accessible and low-cost, which is not.
2. The “2. Lipid and Water molecules in a magnetic field” paragraph and the “2. MRI methodologies” share the same number. Should they be separated? Moreover, in these two paragraphs there is redundancy in the explanation of physical principles, so I suggest a summary, perhaps only in one paragraph.
3. Page 4; lines 199-200 “MRI scans can reveal the presence of setae”, please correct “setae”.
4. Page 4; line 209 “STIR acquisitions to detect bone abnormalities” should be modified in bone marrow abnormalities.
5. Page 5; line 222, in my opinion “The fact that fat and water molecules resonate at…”sounds slightly cacophonous, so I suggest just to remove “The fact that”
6. Page 8; line 364, “The software used for T2 calculation is [42] while…” probably it is a typo, so I suggest adding the name of the software used for T2 calculations.
Overall, the manuscript is readable, although the use of English needs to be revised to try to make the text more fluent. The proposed image is very useful, showing an interesting case of hibernoma, which is suitable to show the potential of T2 relaxometry and fat fraction analysis. In general, I would have added a few more example images to give more weight to the role of T1-DIXONs. The conclusion paragraph is well structured, stressing the potential of quantitative and functional analysis of adipose tissue, which nowadays acquires more and more dignity as an endocrine organ. Comments on the Quality of English Language

language is not addressed

Author Response

Thank you very much for your comments. Appart from editing of the English language, and the addition of a "limitations" section, we also made the following changes:

Comment  1:  "Relatively cheap" was changed to at reasonable cost

Comment 2: The section numbering has been corrected.

Comment 3: "Setae" has been corrected to "septa"

Comment 4: “STIR acquisitions to detect bone abnormalities” was modified to "bone marrow abnormalities".

Comment 5: “The fact that fat and water molecules resonate at…”was changed to "The chemical shift phenomenon between fat and water"

Comment 6:  T2 map is calculated by an in-house built software, we added this information in the text.

Reviewer 2 Report

Comments and Suggestions for Authors

MRI methods to visualize and quantify adipose tissue in health and disease

MRI is becoming more frequently used in studies involving measurements of adipose tissue and volume and composition of skeletal muscles. Topic is very important for diagnosis and treatment of metabolic syndrome. References must be up-dated.

The endocrine, metabolic and hematological functions of fat demand for imaging meth- 381 ods to evolve and become sensitive to highlight and characterize pathology and physiol- 382 ogy in a manner that is repeatable accurate and robust. MR methodologies and related 383 biomarkers play a very significant role in presenting macroscopic fat depots, in detecting 384 microscopic fatty infiltration and in molecularly describing fat content. 

Author Response

Please, find a revised version of the manuscript. It has been edited for English language and a "Limitations" section has been added. Thenk you very much for reviewing this work.

Reviewer 3 Report

Comments and Suggestions for Authors

The manuscript entitled - MRI methods to visualize and quantify adipose tissue in health and disease by Nikiforaki K et al., is an interesting work.  Really gives a deep insight into the applicability of MRI in obesity and associated complications using adipose tissue as target. 

There are no major concerns at this stage. The minor comments are mentioned below.

1 . Kindly perform a proof reading to the revised version.

2 . Include a small section before conclusion section describing the pitfalls or limitations of the work.

Comments on the Quality of English Language

Only minor edits and proof reading required.

Author Response

Thank you very much for your review. We would lie to inform you that a "Limitations" section has been added. Moreover the work has been reviewed for english language.

Round 2

Reviewer 1 Report

Comments and Suggestions for Authors

The authors have clarified all the questions I raised in my previous review. 

The paragraphs are more fluent and well-organized.

Typos have been corrected. I really appreciated the addition of the "limitations" paragraph. In my opinion, as it stands, the article is well written and ready for approval.